# A Unified Feature Disentangler for Multi-Domain Image Translation and Manipulation

**Alexander H. Liu**[1]    **Yen-Cheng Liu**[2]    **Yu-Ying Yeh**[3]    **Yu-Chiang Frank Wang**[1,4]

[1]National Taiwan University, Taiwan
[2]Georgia Institute of Technology, USA    [3]University of California, San Diego, USA
[4]MOST Joint Research Center for AI Technology and All Vista Healthcare, Taiwan

b03902034@ntu.edu.tw, ycliu@gatech.edu
yuyeh@eng.ucsd.edu, ycwang@ntu.edu.tw

## Abstract

We present a novel and unified deep learning framework which is capable of learning domain-invariant representation from data across multiple domains. Realized by adversarial training with additional ability to exploit domain-specific information, the proposed network is able to perform *continuous* cross-domain image translation and manipulation, and produces desirable output images accordingly. In addition, the resulting feature representation exhibits superior performance of unsupervised domain adaptation, which also verifies the effectiveness of the proposed model in learning disentangled features for describing cross-domain data.

## 1   Introduction

Learning interpretable feature representation has been an active research topic in the fields of computer vision and machine learning. In particular, learning deep representation with the ability to exploit relationship between data across different data domains has attracted the attention from the researchers. Recent developments of deep learning technologies have shown progress in the tasks of cross-domain visual classification [6, 26, 27] and cross-domain image translation [10, 24, 30, 11, 28, 17, 16, 4]. While such tasks typically learn feature mapping from one domain to another or derive a joint representation across domains, the developed models have limited capacities in manipulating specific feature attributes for recovering cross-domain data.

With the goal of understanding and describing underlying explanatory factors across distinct data domains, cross-domain representation disentanglement aims to derive a joint latent feature space, where selected feature dimensions would represent particular semantic information [1]. Once such a disentangled representation across domains is learned, one can describe and manipulate the attribute of interest for data in either domain accordingly. While recent work [18] have demonstrated promising ability in the above task, designs of exisitng models typically require high computational costs when more than two data domains or multiple feature attributes are of interest.

To perform joint feature disentanglement and translation across multiple data domains, we propose a compact yet effective model of *Unified Feature Disentanglement Network* (UFDN), which is composed of a pair of unified encoder and generator as shown in Figure 1. From this figure, it can be seen that our encoder takes data instances from multiple domains as inputs, and a domain-invariant latent feature space is derived via adversarial training, followed by a generator/decoder which recovers or translates data across domains. Our model is able to disentangle the underlying factors which represent domain-specific information (e.g., domain code, attribute of interest, etc.). This is achieved

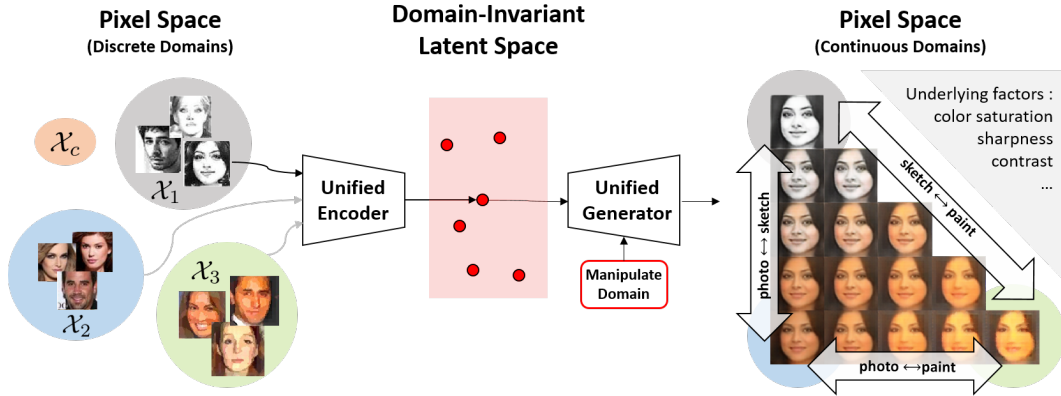

Figure 1: Illustration of multi-domain image translation and manipulation. With data from different domains (e.g., $D_1$: sketch, $D_2$: photo, $D_3$: painting), the goal is to learn domain-invariant feature representation. With domain information disentangled from such representation, one can synthesize and manipulate image outputs in different domains of interests (including the intermediate ones across domains).

by joint learning of our generator. Once the disentangled domain factors are observed, one can simply synthesize and manipulate the images of interest as outputs.

Later in the experiments, we show that the use of our derived latent representation achieves significant improvements over state-of-the-art methods in the task of unsupervised domain adaptation. In addition to very promising results in multi-domain image-to-image translation, we further confirm that our UFDN is able to perform *continuous* image translation using the interpolated domain code in the resulting latent space. Implementation of our proposed method and the datasets are now available[1].

The contributions of this paper are highlighted as follows:

- We propose a Unified Feature Disentanglement Network (UFDN), which learns deep disentangled feature representation for multi-domain image translation and manipulation.

- Our UFDN views both data domains and image attributes of interest as latent factors to be disentangled, which realizes multi-domain image translation in a single unified framework.

- Continuous multi-domain image translation and manipulation can be performed using our UFDN, while the disentangled feature representation shows promising ability in cross-domain classification tasks.

## 2 Related Work

**Representation Disentanglement**   Based on the development of generative models like generative adversarial networks (GANs) [8, 21] and variational autoencoders (VAEs) [13, 22], recent works on representation disentangling [20, 9, 3, 14, 12, 18] aim at learning an interpretable representation using deep neural networks with different degrees of supervision. In a fully supervised setting, Kulkarni *et al.* [14] learned invertible graphic codes for 3D image rendering. Odena *et al.* [20] achieved representation disentanglement with the proposed auxiliary classifier GAN (AC-GAN). Kingma *et al.* [12] also extended VAE into semi-supervised setting for representation disentanglement. Without utilizing any supervised data, Chen *et al.* [3] decomposed representation by maximizing the mutual information between the latent factors and the synthesized images. Despite promising performances, the above works focused on learning disentangled representation of images in a single domain, and they cannot be easily extended to describe cross-domain data. While a recent work by Liu *et al.* [18] addressed cross-domain disentangled representation with only supervision from single-domain data, empirical studies were performed to determine their network architecture (i.e.,

---

number of sharing layers across domains), which would limit its practical uses. Thus, a unified disentangled representation model (like ours) for describing and manipulating multi-domains data would be desirable.

**Image-to-Image Translation** Image-to-image translation is another line of research to deal with cross-domain visual data. With the goal of translating images across different domains, Isola *et al.* [10] applied conditional GAN which is trained on pairwise data across source and target domains. Taigman *et al.* [24] removed the restriction of pairwise training images and presented a Domain Transfer Network (DTN) which observes cross-domain feature consistency. Likewise, Zhu *et al.* [30] employed a cycle consistency loss in the pixel space to achieve unpaired image translation. Similar ideas were applied by Kim *et al.* [11] and Yi *et al.* [28]. Liu *et al.* [17] presented coupled GANs (CoGAN) with sharing weight on high-level layers to learn the joint distribution across domains. To achieve image-to-image translation, they further integrated CoGAN with two parallel encoders [16]. Nevertheless, the above dual-domains models cannot be easily extended to multi-domain image translation without increasing the computation costs. Although Choi *et al.* [4] recently proposed an unified model to achieve multi-domain image-to-image translation, their model does not exhibit ability in learning and disentangling desirable latent representations (as ours does).

**Unsupervised Domain Adaptation (UDA)** Unsupervised domain adaptation (UDA) aims at classifying samples in the target domain, using labeled and unlabeled training data in source and target domains, respectively. Inspired by the idea of adversarial learning [8], Ganin *et al.* [6] proposed a method applying adversarial training between domain discriminator and normal convolution neural network based classifier, making the model invariant to the domain shift. Tzeng *et al.* [26] also attempted to build domain-invariant classifier via introducing domain confusion loss. By advancing adversarial learning strategies, Bousmalis *et al.* [2] chose to learn orthogonal representations, derived by shared and domain-specific encoders, respectively. Tzeng *et al.* [27] addressed UDA by adapting CNN feature extractors/classifier across source and target domains via adversarial training. However, the above methods generally address domain adaptation by eliminating domain biases. There is no guarantee that the derived representation would preserve semantic information (e.g., domain or attribute of interest). Moreover, since the goal of UDA is visual classification, image translation (dual or multi-domains) cannot be easily achieved. As we highlighted in Sect. 1, our UFDN learns the multi-domain disentangled representation, which enables multi-domain image-to-image translation and manipulation and unsupervised domain adaption. Thus, our proposed model is very unique.

## 3 Unified Feature Disentanglement Network

We present a unique and unified network architecture, Unified Feature Disentanglement Network (UFDN), which disentangles the domain information from latent space and derives domain-invariant representation from data across multiple domains (not just from a pair of domains). This not only enables the task of multi-domain image translation/manipulation, the derived feature representation can also be applied for unsupervised domain adaptation.

Given image sets $\{\mathcal{X}_c\}_{c=1}^N$ across $N$ domains, our UFDN learns a domain-invariant representation $z$ for the input image $x_c \in \mathcal{X}_c$ (in domain $c$). This is realized by disentangling the domain information in the latent space as domain vector $v \in \mathbb{R}^N$ via self-supervised feature disentanglement (Sect. 3.1), followed by preserving the data recovery ability via adversarial learning in the pixel space (Sect. 3.2). We now detail our proposed model.

### 3.1 Self-supervised feature disentanglement

To learn disentangled representation across data domains, one can simply apply a VAE architecture (e.g., components $E$ and $G$ in Figure 2). To be more specific, we have encoder $E$ take the image $x_c$ as input and derive its representation $z$, which is combined with its domain vector $v_c$ to reconstruct the image $\hat{x}_c$ via Generator $G$. Thus, the objective function of VAE is defined as:

$$\mathcal{L}_{vae} = \|\hat{x}_c - x_c\|_F^2 + KL(q(z|x_c)||p(z)), \tag{1}$$

where the first term aims at recovering the synthesized output in the same domain $c$, and the second term calculates *Kullback-Leibler divergence* which penalizes deviation of latent feature from the prior

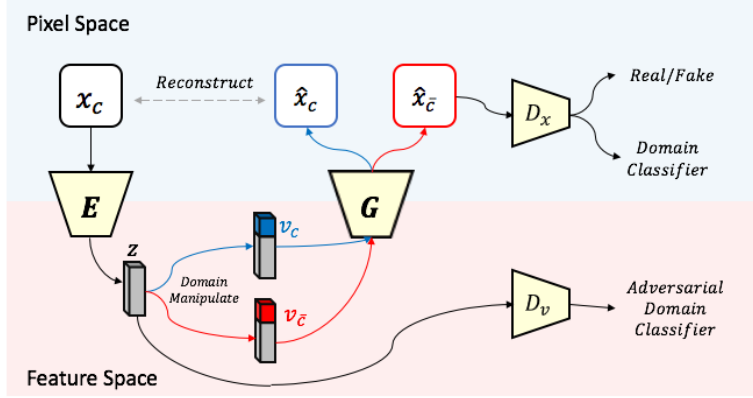

Figure 2: Overview of our Unified Feature Disentanglement Network (UFDN), consisting of an encoder $E$, a generator $G$, a discriminator in pixel space $D_x$ and a discriminator in feature space $D_v$. Note that $x_c$ and $\hat{x}_c$ denote input and reconstruct images with domain vector $v_c$, respectively. $\hat{x}_{\bar{c}}$ indicates the synthesized image with domain vector $v_{\bar{c}}$.

distribution $p(z_c)$ (as $z \sim \mathcal{N}(0, I)$). However, the above technical is not guaranteed to disentangle domain information from the latent space, since generator recovers the images simply based on the representation $z$ without considering the domain information.

To address the above problem, we extend the aforementioned model to eliminate the domain-specific information from the representation $z$. This is achieved by exploiting adversarial domain classification in the resulting latent feature space. More precisely, the introduced domain discriminator $D_v$ in Figure 2 only takes the latent representation $z$ as input and produce domain code prediction $l_v$. The objective function of this domain discriminator $\mathcal{L}_{D_v}^{adv}$ is derived as follows:

$$\mathcal{L}_{D_v}^{adv} = \mathbb{E}[\log P(l_v = v_c | E(x_c))], \tag{2}$$

where $P$ is the probability distribution over domains $l_v$, which is produced by the domain discriminator $D_v$. The domain vector $v_c$ can be implemented by an one-hot vector, concatenation of multiple one-hot vectors, or simply a real-value vector describing the domain of interest. In contrast, the encoder $E$ aims to confuse $D_v$ from correctly predicting the domain code. As a result, the objective of the encoder $\mathcal{L}_E^{adv}$ is to maximize the entropy of the domain discriminator:

$$\mathcal{L}_E^{adv} = -\mathcal{L}_{D_v}^{adv} = -\mathbb{E}[\log P(l_v = v_c | E(x_c))]. \tag{3}$$

## 3.2 Adversarial learning in pixel space

Once the above domain-invariant representation $z$ is learned, we further utilize the reconstruction module in our UFDN to preserve the recovery ability of the disentangled representation. That is, the reconstructed image $\hat{x}_c$ can be supervised by its original image $x_c$.

However, when manipulating the domain vector as $v_{\bar{c}}$ in the above process, there is no guarantee that the synthesized image $\hat{x}_{\bar{c}}$ could be practically satisfactory based on $v_{\bar{c}}$. This is due to the fact that there is no pairwise training data (i.e., $x_c$ and $x_{\bar{c}}$) to supervise the synthesized image $\hat{x}_{\bar{c}}$ in the training stage. Moreover, as noted in [29], the VAE architecture tends to generate blurry samples, which would not be desirable for practical uses.

To overcome the above limitation, we additionally introduce an image discriminator $D_x$ in the pixel space for our UFDN. This discriminator not only improves the image quality of the synthesized image $\hat{x}_{\bar{c}}$, it also enhances the ability of disentangling domain information from the latent space. We note that the objectives of this image discriminator $D_x$ are twofold: to distinguish whether the input image is real or fake, and to predict the observed images (i.e., $\hat{x}_{\bar{c}}$ and $x_c$) into proper domain code/categories.

Table 1: Comparisons with recent works on image-to-image translation.

| | Unpaired data | Bidirectional translation | Unified structure | Multiple domains | Joint representation | Feature disentanglement |
|---|---|---|---|---|---|---|
| Pix2Pix [10] | - | - | - | - | - | - |
| CycleGAN [30] | ✓ | ✓ | - | - | - | - |
| StarGAN [4] | ✓ | ✓ | ✓ | ✓ | - | - |
| DTN [24] | ✓ | - | - | - | ✓ | - |
| UNIT [16] | ✓ | ✓ | - | - | ✓ | - |
| E-CDRD [18] | ✓ | ✓ | - | ✓ | ✓ | ✓ |
| UFDN (Ours) | ✓ | ✓ | ✓ | ✓ | ✓ | ✓ |

With the above discussions, we define the objective functions $\mathcal{L}_{D_x}^{adv}$ and $\mathcal{L}_{G}^{adv}$ for adversarial learning between image discriminator $D_x$ and generator $G$ as:

$$\begin{aligned}
\mathcal{L}_{D_x}^{adv} &= \mathbb{E}[\log(D_x(\hat{x}_{\bar{c}}))] + \mathbb{E}[\log(1 - D_x(x_c)], \\
\mathcal{L}_{G}^{adv} &= -\mathbb{E}[\log(D_x(\hat{x}_{\bar{c}}))].
\end{aligned} \tag{4}$$

On the other hand, the objective function for domain classification is derived as:

$$\mathcal{L}_{cls} = \mathbb{E}[\log P(l_x = v_{\bar{c}}|\hat{x}_{\bar{c}})] + \mathbb{E}[\log P(l_x = v_c|x_c)], \tag{5}$$

where $l_x$ denotes the domain prediction of image discriminator $D_x$. This term implicitly maximizes the mutual information between the domain vector and the synthesized image [3].

To train our UFDN, we alternately update encoder $E$, generator $G$, domain discriminator $D_v$, and image discriminator $D_x$ with the following gradients:

$$\begin{aligned}
\theta_E &\xleftarrow{+} -\Delta_{\theta_E}(\mathcal{L}_{vae} + \mathcal{L}_{E}^{adv}), & \theta_G &\xleftarrow{+} -\Delta_{\theta_G}(\mathcal{L}_{vae} + \mathcal{L}_{G}^{adv} + \mathcal{L}_{cls}), \\
\theta_{D_v} &\xleftarrow{+} -\Delta_{\theta_{D_v}}(\mathcal{L}_{D_v}^{adv}), & \theta_{D_x} &\xleftarrow{+} -\Delta_{\theta_{D_x}}(\mathcal{L}_{D_x}^{adv} + \mathcal{L}_{cls}).
\end{aligned} \tag{6}$$

### 3.3 Comparison with state-of-the-art cross-domain visual tasks

To demonstrate the uniqueness of our proposed UFDN, we compare our model with several state-of-the-art image–to–image translation works in Table 1.

Without the need of pairwise training data, CycleGAN [10] learns bidirectional mapping between two pixel spaces, while they needed to learn the multiple individual networks for the task of multi-domain image translation. StarGAN [4] alleviates the above problem by learning a unified structure. However, it does not exhibit the ability to disentangle particular semantics across different domains. Another line of works on image translation is to learn a joint representation across image domains [24, 16, 18]. While DTN [24] learns a joint representation to translate the image from one domain to another, their model only allows the task of unidirectional image translation. UNIT [16] addresses the above problem by jointly synthesizing the images in both domains. However, it is not able to learn disentangled representation as ours does. A recent work of E-CDRD [18] derives cross-domain representation disentanglement. Their model requires high computational costs when more than two data domains are of interest, while ours is a unified architecture for multiple data domains (i.e., domain code as a vector).

It is worth repeating that our UFDN does not require pairwise training data for learning multi-domain disentangled feature representation. As verified later in the experiments, our model not only enables multi-domain image-to-image translation and manipulation, the derived domain-invariant feature further allows unsupervised domain adaptation.

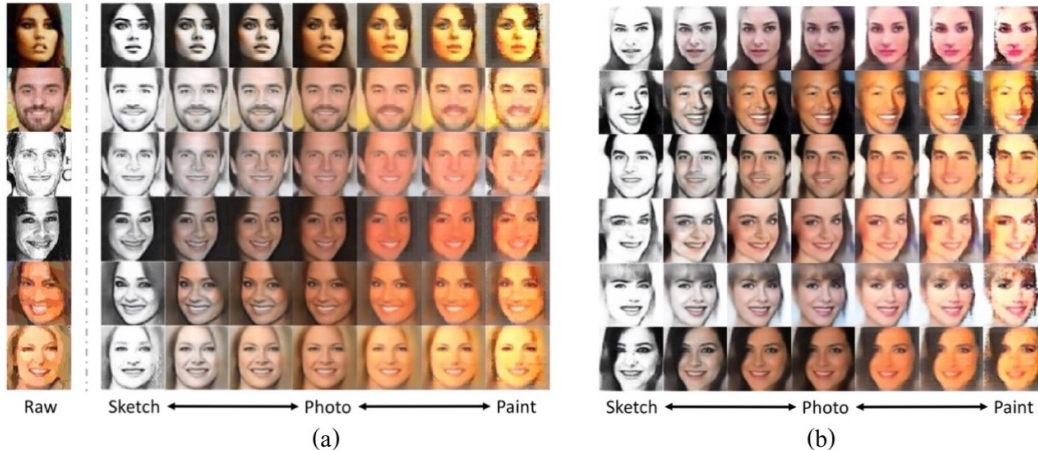

Figure 3: (a) Examples results of image-to-image translation across data domains of sketch/photo/paint and (b) example image translation results with randomly generated identity (i.e., random sample $z$).

## 4 Experiment Results

### 4.1 Datasets

**Digits** *MNIST, USPS, Street View House Number (SVHN)* datasets are considered to be three different domains and used as benchmark datasets in unsupervised domain adaption (UDA) tasks. MNIST contains 60000/10000 training/testing images, and USPS contains 7291/2007 training/testing images. While the above two datasets are handwritten digits, SVHN consists of digit images with the complex background and various illuminations. We used the 60000 images from SVHN extra training set to train our model and few samples from the testing set to perform image translation. All images are converted to RGB images with the size 32x32 in our experiments.

**Human faces** We use the Large-scale CelebFaces Attributes (CelebA) Dataset [19] in our experiment on human face images. CelebA includes more than 200k celebrity photos annotated with 40 facial attributes. Considering *photo*, *sketch* and *paint* as three different domains, we follow the setting of previous works [10, 18] to transfer half of the photos to sketch. We further transferred half of the remaining photos to paint through off-the-shelf style transfer software[2].

### 4.2 Multi-domain image translation with disentangled representation

Most of the previous works focus on image translation between two domains as mentioned in Section 2. In our experiment, we use human face images from different domains to perform image-to-image translation. Although Choi *et al.* [4] claim to have achieved multi-domain image-to-image translation on human face dataset, they define *attribute*, e.g., gender or hair color, as *domain*. In our work, we denote *domain* by the dataset properties rather than attributes. Images from different domains may share same attributes, but an image cannot belong to two domain at the same time.

With unified framework and no restriction on the dimension of domain vector, UFDN can perform image-to-image translation over multiple domains. As shown in Figure 3(a), we demonstrate the results of image-to-image translations between domains *sketch/photo/paint*. Previous works [25, 15] had discovered that even the disentangled feature to the generator/decoder is binary during training, it can be considered as continuous variable during testing. Our model also inherits this property of continuous cross-domain image translation by manipulating the value of domain vector.

Our model is also capable of generating unseen images by randomly sampled representation in the latent space. Since the representation is sampled from domain-invariant latent space, UFDN can further present them with any domain vector supplied. Figure 3(b) shows the result of translation for

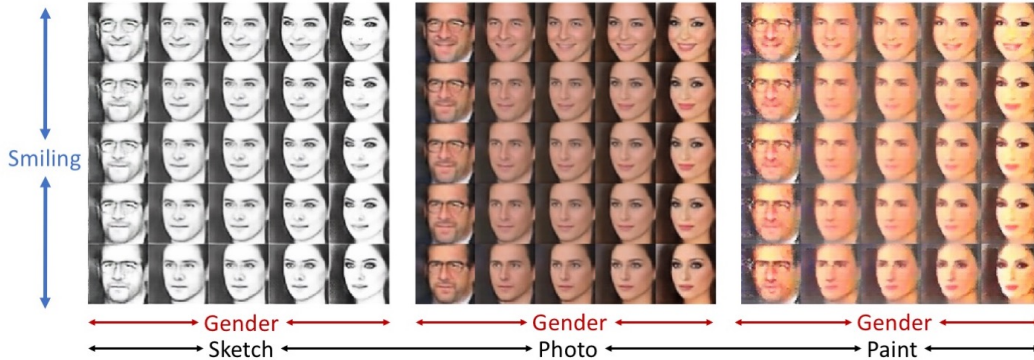

Figure 4: Example results of multi-domain image translation. Note that all images are produced by the same $z$ with varying domain information.

Table 2: Quantitative evaluation in terms of image-to-image translation on human face dataset.

|  | Sketch→Photo | | | Paint→Photo | | |
|---|---|---|---|---|---|---|
|  | SSIM | MSE | PSNR | SSIM | MSE | PSNR |
| E-CDRD [18] | 0.6229 | 0.0207 | 16.86 | 0.5892 | 0.0174 | 17.61 |
| StarGAN [4] | 0.8026 | 0.0142 | 19.04 | 0.8496 | 0.0060 | 22.53 |
| UFDN (Ours) | **0.8222** | **0.0106** | **20.24** | **0.8798** | **0.0033** | **25.06** |

six identities randomly sampled. It is worth noting that this cannot be done by those translation models without representation learning or using skipped connection between encoder and decoder/generator.

Table 2 provides quantitative evaluation on the recovered images using our proposed UFDN with E-CDRD [18] and StarGAN[3] [4]. In our experiments, we convert photo images into sketches/paintings for the purpose of collecting training cross-domain image data (but did not utilize such pairwise information during training). This is the reason why we are able to observe the ground truth photo images and calculate SSIM/MSE/PSNR values for the translated outputs. While both learning disentangled representation, our UFDN outperformed E-CDRD in terms translation quality. It is also worth noting that our UFDN matched the performance of StarGAN, which was designed for image translation without learning any representation.

To further demonstrate the ability to disentangle representation, our model performs feature disentanglement of common attributes across domains simultaneously. This can be easily done by expanding domain vector with the annotated attribute from the dataset. In our experiment, *Gender* and *Smiling* are picked as the attribute of interest. The results are shown in the Figure 4. We used a fixed domain-invariant representation to show that features are highly disentangled by our UFDN. All information of interest (domain/gender/smiling) can be independently manipulated through our model. As a reminder, each and every result provided above was presented by the same *single model*.

### 4.3 Unsupervised domain adaption with domain-invariant representation

Unsupervised domain adaption (UDA) aims to classify samples in target domain while labels are only available in the source domain. Previous works [6, 27] dedicated to building a domain-invariant classifier for UDA task. Recent works [16, 18] addressed the problem by using classifier with high-level layers tied across domain and synthesized training data provided by image-to-image translation. We followed the previous works to challenge UDA task on digit classification over three datasets MNIST/USPS/SVHN. The notation "→" denotes the relation between source and target domain. For example, SVHN→MNIST indicates that SVHN is the source domain with categorical labels.

To verify the robustness of our domain-invariant representation, we adapt our model to UDA task by adding a *single fully-connected layer* as the digit classifier. This classifier simply takes as input the

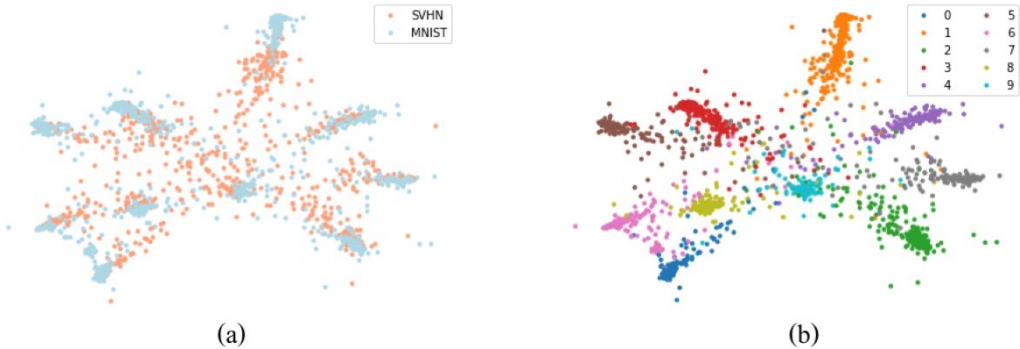

(a)                                                          (b)

Figure 5: t-SNE visualization of SVHN→MNIST. Note that different colors indicate data of (a) different domains and (b) digit classes.

Table 3: Performance comparisons of unsupervised domain adaptation (i.e., classification accuracy for target-domain data). For example, MNIST→USPS denotes MNIST and USPS as source and target-domain data, respectively.

|  | MNIST→USPS | USPS→MNIST | SVHN→MNIST |
|---|---|---|---|
| SA [5] | 67.78 | 48.80 | 59.32 |
| DANN [6] | - | - | 73.85 |
| DTN [24] | - | - | 84.88 |
| DRCN [7] | 91.8 | 73.7 | 82.00 |
| CoGAN [17] | 95.65 | 93.15 | - |
| ADDA [27] | 89.40 | 90.10 | 76.00 |
| UNIT [16] | 95.97 | 93.58 | 90.53 |
| ADGAN [23] | 92.80 | 90.80 | 92.40 |
| CDRD [18] | 95.05 | 94.35 | - |
| **UFDN (Ours)** | **97.13** | 93.77 | **95.01** |

domain-invariant representation and predicts the digit label. The auxiliary classifier is jointly trained with our UFDN.

Table 3 lists and compares the performance of our model to others. For the setting MNIST→USPS, our model surpasses UNIT [16] which was the state-of-the-art. For SVHN→MNIST, our model also surpasses the state-of-the-art with significant improvement. While SVHN→MNIST is considered to be much more difficult than the other two settings, our model is able to decrease the classification error rate from 7.6% to 5%. It is also worth mentioning that our model used 60K images from SVHN, which is considerably less than 531K used by UNIT.

We visualize domain-invariant representations with t-SNE and show the results in Figure 5. From Figure 5(a) and 5(b) we can see that the representation is properly clustered with respect to class of digits instead of domain. We also provide the result of synthesizing images with the domain-invariant representation. As shown in Figure 6, by manipulating domain vector, the representation of SVHN image can be transformed to MNIST. It further strengthens our point of view that disentangled representation is worth learning.

### 4.4 Ablation study

As mentioned in Section 3, we applied self-supervised feature disentanglement and adversarial learning in pixel space to build our framework. To verify the effect of these methods, we did ablation study on the proposed framework and show the results in Figure 7. We claimed that without self-supervised feature disentanglement, i.e., without $D_v$, the generator will be able to reconstruct images with the entangled representation and ignore the domain vector. This can be verified by Figure 7(a) where the self-supervised feature disentanglement is disabled, meaning that the representation is not trained to be domain-invariant. In such case, the decoder simply decodes the input representation back

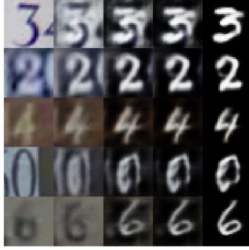

Figure 6: Example image translation results of SVHN→MNIST.

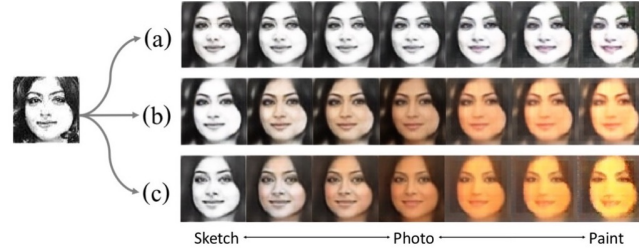

Figure 7: Comparison between example image translation results: (a) our UFDN without self-supervised feature disentanglement, (b) our UFDN without adversarial training in the pixel space, and (c) the full version of UFDN.

to its source domain ignoring the domain vector. Next, we disabled pixel space adversarial learning in our framework to verify that the representation is indeed forced to be domain-invariant. As shown in Figure 7(b), the generator is now forced to synthesize image conditioning on the manipulated domain vector. However, without pixel space adversarial learning, the difference between domain *photo* and *paint* is not apparent comparing to the complete version of our UFDN.

## 5  Conclusion

We proposed a novel network architecture of unified feature disentanglement network (UFDN), which learns disentangled feature representation for data across multiple domains by a unique encoder-generator architecture with adversarial learning. With superior properties over recent image translation works, our model not only produced promising qualitative results but also allows unsupervised domain adaptation, which confirmed the effectiveness of the derived deep features in the above tasks.

## Footnotes

[2]`https://fotosketcher.com`

[3]We used the source code provided by the author at `https://github.com/yunjey/StarGAN`

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
