[Supplementary Material]

# A   Network architecture & training details

**Human face translation**   The network architecture used for image translation between *sketch*, *photo* and *paint* is given in Table A.

Encoder/Decoder utilizes either convolutional layer or transposed convolutional layer, and Discriminator utilizes convolutional layer or fully connected layer (FC). We note that the domain vector $v$ is a 3 dimensional one-hot vector combined with a 4 dimensional binary vector (2 binary bits for each attribute) for the attribute of interest. (7 dimensions in total)

During training, we employ ADAM optimizer with $\beta_1 = 0.5$ and $\beta_2 = 0.99$. The learning rate is set as $1e-4$, and the batch size is set as 96 (32 from each domain). We apply WGAN-GP as our pixel space discriminator. The weight for the objective in Equation 6 is set as follow: $2e-4$ for *KL divergence* in $\mathcal{L}_{vae}$, 0.01 for $\mathcal{L}_G^{adv}$ and $\mathcal{L}_{cls}$, and 1 for others. *KL divergence* is ignored in the experiment of Table 2 since generating unseen samples is not necessary.

Table A: The network architecture of UFDN for *sketch*, *photo* and *paint*.

| Layer | Filters | Kernel size | Stride | BatchNorm. | Activation |
|---|---|---|---|---|---|
| **Encoder** | | | | | |
| **Input : 64x64x3** | | | | | |
| 1 | 64 | 4x4 | 2 | ✓ | Leaky ReLU |
| 2 | 128 | 4x4 | 2 | ✓ | Leaky ReLU |
| 3 | 256 | 4x4 | 2 | ✓ | Leaky ReLU |
| 4 | 512 | 4x4 | 2 | ✓ | Leaky ReLU |
| 5 | 1024 | 4x4 | 2 | ✓ | Leaky ReLU |
| $\mu$ | 1024 | 4x4 | 2 | - | - |
| **Generator** | | | | | |
| **Input : 1024+7** | | | | | |
| 1 | 1024 | 4x4 | 2 | ✓ | Leaky ReLU |
| 2 | 512 | 4x4 | 2 | ✓ | Leaky ReLU |
| 3 | 256 | 4x4 | 2 | ✓ | Leaky ReLU |
| 4 | 128 | 4x4 | 2 | ✓ | Leaky ReLU |
| 5 | 64 | 4x4 | 2 | ✓ | Leaky ReLU |
| 6 | 3 | 4x4 | 2 | - | Tanh |
| **Discriminator** | | | | | |
| **Input : 64x64x3** | | | | | |
| 1 | 16 | 4x4 | 2 | - | Leaky ReLU |
| 2 | 32 | 4x4 | 2 | - | Leaky ReLU |
| 3 | 64 | 4x4 | 2 | - | Leaky ReLU |
| 4 | 128 | 4x4 | 2 | - | Leaky ReLU |
| 5 (FC) | 512 | - | - | - | Leaky ReLU |
| 6 (FC) | [1,7] | - | - | - | - |

**UDA on digits**   The network architecture and training hyper-parameters used for *MNIST→USPS* and *USPS→MNIST* are identical as illustrated in Table B.

Encoder/Decoder uses convolutional layer/transposed convolutional layer. The digit classifier is a single layer fully-connected network which is jointly learned with our UFDN (only labels in source domain). We note that the domain vector $v$ is a 2 dimensional one-hot vector. we employ ADAM optimizer with $\beta_1 = 0.5$ and $\beta_2 = 0.99$. The learning rate is set as $1e-4$, and the batch size is set as 32 (16 from each domain).

The weight for the objective in Equation 6 is set as follow: $1e-7$ for *KL divergence* in $\mathcal{L}_{vae}$, 0.1 for $\mathcal{L}_E^{adv}$ and 1 for others. In the task of UDA, we discover that the pixel space discriminator $D_x$ is not necessary since we perform classification on domain-invariant representations instead of images.

The network architecture used for UDA on *SVHN→MNIST* is given in Table C.

Table B: The network architecture of UFDN for *MNIST→USPS* and *USPS→MNIST*.

| | | **Encoder** | | | |
| | | Input : 32x32x3 | | | |
| Layer | Filters | Kernel size | Stride | BatchNorm. | Activation |
|---|---|---|---|---|---|
| 1 | 64 | 4x4 | 2 | ✓ | Leaky ReLU |
| 2 | 128 | 4x4 | 2 | ✓ | Leaky ReLU |
| 3 | 256 | 4x4 | 2 | ✓ | Leaky ReLU |
| 4 | 512 | 4x4 | 2 | ✓ | Leaky ReLU |
| $\mu$ | 1024 | 4x4 | 2 | - | - |
| | | **Generator** | | | |
| | | Input : 64+2 | | | |
| Layer | Filters | Kernel size | Stride | BatchNorm. | Activation |
| 1 | 512 | 4x4 | 2 | ✓ | Leaky ReLU |
| 2 | 256 | 4x4 | 2 | ✓ | Leaky ReLU |
| 3 | 128 | 4x4 | 2 | ✓ | Leaky ReLU |
| 4 | 64 | 4x4 | 2 | ✓ | Leaky ReLU |
| 5 | 3 | 4x4 | 2 | - | Tanh |
| | | **Digit Classifier** | | | |
| | | Input : 64 | | | |
| Layer | Filters | Kernel size | Stride | BatchNorm. | Activation |
| 1 (FC) | 10 | - | - | - | Soft-max |

Encoder/Decoder utilizes either convolutionalal layer or transposed convolutionalal layer, and the digit classifier is a single layer fully-connected network that is jointly learned with our UFDN (with labels in source domain). We note that the domain vector $v$ is a 2 dimensional one-hot vector.

During training, we employ ADAM optimizer with $\beta_1 = 0.5$ and $\beta_2 = 0.99$. The learning rate is set as $1e - 4$, and the batch size is set as 32 (16 from each domain).

Table C: The network architecture of UFDN for *SVHN →MNIST*.

| | | **Encoder** | | | |
| | | Input : 32x32x3 | | | |
| Layer | Filters | Kernel size | Stride | BatchNorm. | Activation |
|---|---|---|---|---|---|
| 1 | 128 | 4x4 | 2 | ✓ | Leaky ReLU |
| 2 | 256 | 4x4 | 2 | ✓ | Leaky ReLU |
| 3 | 512 | 4x4 | 2 | ✓ | Leaky ReLU |
| 4 | 1024 | 4x4 | 2 | ✓ | Leaky ReLU |
| $\mu$ | 2048 | 4x4 | 2 | - | - |
| | | **Generator** | | | |
| | | Input : 2048+2 | | | |
| Layer | Filters | Kernel size | Stride | BatchNorm. | Activation |
| 1 | 1024 | 4x4 | 2 | ✓ | Leaky ReLU |
| 2 | 512 | 4x4 | 2 | ✓ | Leaky ReLU |
| 3 | 256 | 4x4 | 2 | ✓ | Leaky ReLU |
| 4 | 128 | 4x4 | 2 | ✓ | Leaky ReLU |
| 5 | 3 | 4x4 | 2 | - | Tanh |
| | | **Digit Classifier** | | | |
| | | Input : 2048 | | | |
| Layer | Filters | Kernel size | Stride | BatchNorm. | Activation |
| 1 (FC) | 10 | - | - | - | Soft-max |