[Reviews · NeurIPS 2018]

Reviewer 1



In overall, I think this paper proposes a well-designed unified framework which benefits many applications. The rebuttal addresses my questions and the comparison results are convincing to me. I strongly suggest the authors add these results and discussions in the final version if this paper is accept. I raise my rating to accept. ============================================ This paper proposes a unified multi-domain feature disentangler framework which enables cross-domain image translation, image manipulation, and classification domain adaptation. Pros: * This paper combines all the benefits from previous works into a simple unified framework which can be beneficial to many tasks. * The learned domain-invariant features perform well on the unsupervised domain adaptation task, especially on SVHN → MNIST experiment. Cons: * The domain vector/code is not explained clearly in the main paper. * Since there is paired ground truth for the face experiments, I think it is better to use some quantitative metrics (e.g. multi-channel SSIM) to evaluate the proposed method and compare to other methods (e.g. the close related work [17]) if possible. * As to the face experiment, do the images not overlap between the three domains (sketch/photo/paint) as mentioned in line 181? Given that the proposed method also uses domain classifier to encourage the encoder to learn domain invariant representation, it is suggested to cite the related work, K. Bousmalis, G. Trigeorgis, N. Silberman, D. Krishnan and D. Erhan. “Domain separation networks.” In NIPS, 2016.

Reviewer 2



Paper summary: The authors propose a VAE-GAN architecture for learning multi-domain image representations where the domain information is disentangled from the domain invariant information, while simultaneously generating images of all considered domains. A domain classifier D_v pushes the VAE encoder E to learn a domain-invariant embedding z by adversarial training. This embedding is then concatenated with a one-hot encoding v of the domain before being fed to the VAE decoder G, which is then able to reconstruct domain-appropriate images. The VAE is further paired with an adversarial real / fake classifier, which improves image sharpness, and a domain classifier, which promotes mutual information between the generated images and the domain encoding v. Experiments are performed on human face generation and semi-supervised digits classification. Comments: The proposed approach is pretty straightforward when compared to recent works on image generation / feature disentanglement (which is good), and the "unified" nature of the proposed architecture must be appreciated. The paper is well written and easy to follow, although some technical details are not sufficiently clear from the text (point #2 below). Furthermore, in my opinion the authors tend to overstate their contributions w.r.t existing methods, in particular the one in [17] (point #1 below). Finally, the experimental evaluation on image generation is a bit lacking (point #3 below). My current vote is for rejection, but I'm willing to increase my score given satisfying answers to the following three main points: 1) In my opinion the novelty of the proposed method w.r.t. the one in [17] is quite limited. In fact, the proposed network can be directly derived from the E-CDRD on of [17] by simply merging the source and target branches and adding a domain adversarial classifier to the hidden representation. Can you please elaborate on this? Are there any other differences (theoretical or practical / architectural) that I'm missing? 2) I think that the overall clarity of the paper would benefit from moving part of the information reported in the supplementary material to the main paper. In particular, I find it a bit odd that nowhere in the paper is ever mentioned that v is a one-hot encoding of the domain labels, or that no information is ever given about the architectures of the various modules composing the network. 3) All the results shown on image generation are of a qualitative kind, and there's no comparison with existing methods. While, unfortunately, this is becoming alarmingly common in this line of research, I still think that authors should strive to evaluate as objectively as possible their proposed methods. In fact, it's not inconceivable that the generated images shown in the paper could be carefully selected outliers. Performing this kind of objective evaluation is a non-trivial tasks, but some de-facto standards currently exist, including Mechanical Turk-based perceptual evaluation (like in [3]), Inception score (like in REF1 below), or GAN convergence metrics. In the absence of any of these I would argue that, from the purely scientific point of view, the results in Section 4.2 are effectively meaningless. REF1: Tim Salimans, Ian Goodfellow, Wojciech Zaremba, Vicki Cheung, Alec Radford, and Xi Chen. Improved techniques for training gans. In Neural Information Processing Systems (NIPS), 2016 --- Additional comments after author response phase --- The authors gave satisfactory answers to my three main points of criticism, although I still think that the novelty of the proposed method w.r.t the state of the art is a bit limited and should not be emphasized too much in the text. Overall, I think the paper can be accepted.

Reviewer 3



Paper proposed a joint feature disentanglement and translation across multiple data domains. Architecture consists of an encoder that can encode an image from any of the domains of interest into domain-invariant latent space; and a generator that given a latent space vector and domain can reconstruct the image in that domain. The architecture his a combination of the VAE and adversarial GAN training. In particular, VAE encoder is used to encode an image into the latent space. An adversarial discriminator is trained on the latent codes to ensure that the latent code vectors are not able to distinguish a domain of the input. At the same time latent vector augmented with domain label are propagated through generator that is also trained in adversarial fashion, by ensuring that the generated image is real and is in the “right” domain after the reconstruction. Overall the approach is sensible and sound. Paper is reasonably easy to read and formulation is compelling. However, comparisons to important baselines are missing. In particular, there are a number of recent methods for multi-domain image translation that should be compared against. The best example is StarGAN, which experiments also include pair-wise variants of CycleGAN among others. These should certainly be compared against. Other papers that is probably worth looking at are the following: Unsupervised Multi-Domain Image Translation with Domain-Specific Encoders/Decoders L. Hui, X. Li, J. Chen, H. He, C. Gong, J. Yang and Modular Generative Adversarial Networks B. Zhao, B. Cheng, Z. Jie, L. Sigal Although these were “unpublished” at the time of NIPS submission, so direct comparison might not be necessary. Comparison with StarGAN is however. The argument made for not including such comparison is that authors define “domains” differently, but this is just a matter of semantics. Certainly an experiment under attribute-based definition of StarGAN would be possible and really necessary for validation of the approach.